# Electronic Skin Based on Polydopamine-Modified Superelastic Fibers with Superior Conductivity and Durability

**DOI:** 10.3390/nano14050438

**Published:** 2024-02-28

**Authors:** Chengfeng Chen, Yimiao Wang, Hang Wang, Xinqing Wang, Mingwei Tian

**Affiliations:** 1Intelligent Wearable Engineering Research Center of Qingdao, College of Textiles and Clothing, Qingdao University, Qingdao 266071, China; 2021020769@qdu.edu.cn (C.C.); 2020204909@qdu.edu.cn (Y.W.); 2020204866@qdu.edu.cn (X.W.); 2Shandong Special Nonwoven Materials Engineering Research Center, Qingdao University, Qingdao 266071, China

**Keywords:** flexible sensor, electrospun fibers, hydrogel, interface effect, modification

## Abstract

Owing to their excellent elasticities and adaptability as sensing materials, ionic hydrogels exhibit significant promise in the field of intelligent wearable devices. Nonetheless, molecular chains within the polymer network of hydrogels are susceptible to damage, leading to crack extension. Hence, we drew inspiration from the composite structure of the human dermis to engineer a composite hydrogel, incorporating dopamine-modified elastic fibers as a reinforcement. This approach mitigates crack expansion and augments sensor sensitivity by fostering intermolecular forces between the dopamine on the fibers, the hydrogel backbone, and water molecules. The design of this composite hydrogel elevates its breaking tensile capacity from 35 KJ to 203 KJ, significantly enhancing the fatigue resistance of the hydrogel. Remarkably, its electrical properties endure stability even after 2000 cycles of testing, and it manifests heightened sensitivity compared to conventional hydrogel configurations. This investigation unveils a novel method for crafting composite-structured hydrogels.

## 1. Introduction

Wearable sensors, which are emerging as novel electronic devices, have considerable potential for development across diverse fields, including motion detection [1,2], healthcare [3,4], electronic skin [5,6], soft robotics [7,8], and human–machine interfaces [9,10]. Among these sensors, skin-like wearable sensors may convert mechanical signals to electrical signals by mimicking the softness and adaptability of human skin, which can profoundly affect human–computer interactions and motion monitoring applications [11,12]. Ionic hydrogels, comprising a network of covalently and/or non-covalently crosslinked hydrophilic polymers and solvents, represent promising skin-like wearable sensors. These hydrogels, serving as multifunctional sensing materials, demonstrate remarkable capabilities in bending, stretching, and deformation. In comparison to conventional flexible sensors, hydrogel sensors offer distinct advantages, including adaptability to diverse curved surfaces, lightweight construction, and a broad sensing range. Ion-doped hydrogels exhibit modulus and elasticity akin to human skin, thus holding significant potential in the realms of electronic skin and robotics and garnering considerable attention from researchers [13].

Hydrogels, an elastic substance comprising a polymer matrix and water molecules, derive their physical attributes governing high elasticity and exceptional conformity to the human body. The ion hydrogel, employed in sensor fabrication, features a three-dimensional cross-linked hydrophilic polymer network immersed in a matrix rich in water content. Within these hydrogels, ions manifest fluidity. When subjected to external forces, the gel undergoes deformation, inducing alterations in its electrical signal and thereby realizing sensing functionality. Currently, the preparation methods of hydrogels for strain sensing applications mainly include physical cross-linking [14] or chemical cross-linking [15]. Physical gels materialize through physical phenomena such as electrostatic interactions, hydrogen bonding, and chain entanglement, rendering the internal three-dimensional scaffold susceptible to impairment. Conversely, hydrogels synthesized via chemical cross-linking boast heightened robustness, elasticity, and resilience owing to the presence of covalent links among molecules. Despite the mechanical superiority of ion hydrogels produced through chemical cross-linking over those crafted via physical means, their resilience remains relatively wanting. Particularly under external pressures, the three-dimensional polymer framework of chemically cross-linked hydrogels tends to undergo stress concentration, precipitating irreversible fractures [16] and compromising the structural integrity of their sensing mechanisms. This limitation presents a barrier to meeting the protracted operational requisites of smart wearable technologies. Consequently, in order to fulfill the exigencies of flexible sensors, the optimization of ion hydrogels is imperative to augment the efficacy of hydrogel-based sensors.

Therefore, to satisfy the demands of flexible sensors, the ionic hydrogel should be designed by introducing other substances [17,18] and increasing the toughness of the structure [19,20], etc., to enhance the use of hydrogel sensors. The introduction of other polymer materials or reinforcers into the hydrogel matrix may effectively enhance the comprehensive performance of the hydrogel. To improve the durability of flexible hydrogel sensors, Yang et al. [21] introduced a dual-network hydrogel ionic conductor that combined the physical cross-linking of agar with the chemical cross-linking of polyacrylamide. This innovative approach yielded excellent mechanical properties, with an elongation at break of up to 1600%. Certain researchers have engineered a composite architecture of flexible hydrogel sensors inspired by biomimicry. The human dermal tissue comprises a supple protein matrix and sturdy collagen fiber scaffolding. This intricate arrangement bestows exceptional anti-fatigue toughness upon human skin, allowing it to endure millions of deformation cycles annually. Through the utilization of elastic microfibers to emulate the collagen fiber framework and hydrogels to replicate the protein matrix, the mechanical attributes of the resultant composite flexible sensors are substantially bolstered. Consequently, their flexibility as composite materials is markedly improved. The fatigue resistance of the hydrogel sensor experiences a significant enhancement. Wang et al. [22] successfully fabricated a composite ion sensor by combining a hydrogel with fibers. This blend effectively enhanced the robustness of the hydrogel, increased its fatigue fracture threshold to 2950 J/m^2^, and significantly improved its resistance to crack extension. The prevailing body of research predominantly emphasizes the incorporation of fibers into hydrogel formulations for reinforcement, but the influences of the structural and functional modifications of the fiber surface within ionic gels are notable oversights. The failure to establish an interface between untreated fibers and the hydrogel can significantly diminish the finishing properties of the composites. To further enhance the durability and sensing capabilities of flexible hydrogel sensors, attention can be directed toward fortifying the interfacial effect between the hydrogel and its reinforcement within the realm of composites. Indeed, feeble interfacial interactions between the hydrogel and its reinforcement may precipitate the deterioration of the mechanical properties of the composites, posing a threat to structural integrity. Overcoming this challenge necessitates a pivotal improvement in the surface characteristics of fibers serving as reinforcements, directly contributing to the comprehensive performance of the sensors.

In this study, we developed a hydrogel composite by combining water-based gel with dopamine-modified polyurethane fibers, namely the dopamine-modified polyurethane fiber hydrogel (DPF hydrogel). Initially, we fabricated ultra-fine and exceedingly elastic polyurethane fibers (PFs) through the electrospinning process. These fibers underwent subsequent modification via dopamine, resulting in dopamine-modified polyurethane fibers (DPFs), wherein dopamine intricately anchors itself onto the fiber’s surface. Post-modification, these fibers are intricately combined with the hydrogel precursor, culminating in the ultimate manifestation of the DPF hydrogel, as elucidated in Figure 1a. The inherent functional groups of dopamine augment the interfacial effect between the fiber and hydrogel (as shown in Figure 1b). This composite material, marrying hydrogel nanofibers with hydrogels, seamlessly integrates the functional attributes of a hydrogel with the structural merits of nanofibers [23]. The amalgamated structure of these fibers and hydrogels ingeniously emulate human dermal tissue, conferring notable fatigue resistance and heightened sensitivity and rendering it eminently suited for flexible sensors. Moreover, the tailored modification of these fibers amplifies the overarching performance of the composite material. This amalgamation not only elevates the mechanical properties of the hydrogel, augmenting its modulus and strength but also fortifies its resilience. This substantial enhancement significantly prolongs the sensor’s operational longevity, thereby amplifying its pragmatic worth. Our empirical observations reveal a noteworthy escalation in tensile fracture work, increasing from 35 kJ in the hydrogel to a commendable 203 kJ in the DPF hydrogel. This provides evidence of a remarkable fortification in strength, effectively mitigating crack propagation within the internal fibers. The sensor’s expansive detection range spans 640%, coupled with an impressive stimulus-response time of 50 milliseconds. Functioning as a monitoring instrument, this composite hydrogel sensor exhibits exemplary human–machine performance, proficiently overseeing movements across diverse regions of the human body, thereby underscoring its tremendous potential in practical applications.

## 2. Materials and Methods

### 2.1. Materials

Polyurethane (TPU, Elastollan 1185A) was purchased from BASF(China) Co., Ltd. (Shanghai, China) N, N-dimethylformamide (DMF, ≥99.5%, analytical reagent) and tetrahydrofuran (THF, ≥99.5%, analytical reagent) were supplied by Shanghai Macklin Biochemical Co., Ltd. (Shanghai, China) Dopamine hydrochloride and tris (hydroxymethyl) aminomethane hydrochloride(tris) were supplied by Shanghai Macklin Biochemical Co., Ltd. N,N′-Methylenebisacrylamide and ammonium persulfate were supplied by Beijing InnoChem Science & Technology Co., Ltd. (Beijing, China) N,N,N′,N′-Tetraethylmethylenediamine was supplied by Alfa Aesar (China) Chemical Co., Ltd. (Shanghai, China).

### 2.2. Methods

#### 2.2.1. Preparation of Elastic Fibers

TPU particles were dissolved in a solvent blend of DMF and THF at a weight ratio of 6:4, yielding a TPU concentration of 13%. The resultant TPU solution underwent magnetic agitation at room temperature for 8 h to ensure a homogenous composition. Subsequently, the prepared TPU solution was loaded into a syringe for electrospinning. Electrospinning was carried out with a voltage of 23 kV and a distance of 12 cm between the collector and the syringe needle. The TPU solution was consistently delivered at a rate of 3 mL/h. A receiving roller, clad in aluminum foil and rotating at a speed of 120 rpm, served as the fiber collection apparatus.

#### 2.2.2. Preparation of Dopamine-Treated Fiber Mesh

Dopamine hydrochloride (1 g) and tris (0.6 g) were dissolved in 500 mL of deionized water. The solution was subject to stirring and ultrasonic shaking. Ammonia water was diluted and then gradually introduced into the dopamine hydrochloride solution to adjust the pH to 8.5.

The electrospun membranes were immersed in a dopamine buffer and maintained for 3, 6, and 9 h to produce dopamine-modified elastic-electrospun membranes, and samples DPF-3, DPF-6, and DPF-9 were prepared.

#### 2.2.3. Preparation of Hydrogel Precursor

First, 4 g of acrylamide, 2 g of LiCl, and 0.007 g of methylenebisacrylamide were dissolved in 30 mL of deionized water. The mixture was stirred using a magnetic stirrer for 20 min until it was completely dissolved. Subsequently, 0.003 g of the ammonium sulfate catalyst was added to the solution and stirred for 10 min, followed by the addition of 15 μL of tetramethylethylenediamine. The solution was meticulously stirred to achieve a hydrogel precursor solution.

#### 2.2.4. Preparation of DPF Hydrogel

The fiber mesh was positioned in a mold and securely anchored in the central layer using a chuck. Thereafter, the hydrogel precursor solution was poured onto the fiber mesh to completely submerge it. The assembly was placed in an oven at 90 °C to initiate hydrogel cross-linking and form a polymer skeleton that would envelop the fibers. Through this process, a fiber–hydrogel ion sensor was fabricated.

### 2.3. Characterization

The surface morphology of PF, DPF-3, DPF-6, and DPF-9 was observed using a field emission scanning electron microscope (Regulus8100, Hitachi Scientific Instruments Beijing Co., Ltd, Beijing, China) to analyze the influence of different polydopamine modifications on fiber surface morphology. PF, DPF-3, DPF-6, and DPF-9 were subjected to contact angle measurements using a contact angle tester to assess the impact of polydopamine modification on fiber wettability through the analysis of contact angles. A universal testing system (3382, Shanghai Instron Testing Equipment Trading Co., Ltd, Shanghai, China) was utilized to stretch the hydrogels and DPF hydrogels at a constant strain rate of 5 mm/min, evaluating the tensile properties of thermoplastic polyurethane fiber networks and flexible sensors, including tensile fracture length and the mechanical hysteresis of DPF hydrogels at deformations of 10%, 50%, and 100%. The sensing system of the sensor was tested using a semiconductor characterization system (4200A-SCS was supplied by China Tech Technology Co., Ltd, Nanjing, China) by subjecting the hydrogel and DPF hydrogel stepper to identical rates of stretching, comparing their relative resistance change rates. Resistance was measured while stretching until sensor failure to determine the detection range. Stepwise sensing tests were conducted at 10 s intervals, observing the relative resistance change rate curves of the sensor in static and dynamic states during the stretching cycle. Tensile cycling assessed the cyclic stability of the flexible sensor’s electrical signal at 10% deformation on the stepper. Pulse tests were conducted using the DPF hydrogel. Firstly, a thin layer of ultra-thin polyvinyl alcohol gel (with a certain deformation capacity after drying) was applied to the pulse area on the skin. The sample size was controlled to only cover the area near the pulse, and the hydrogel sensor sample was attached with slight pre-tension to the monitoring area. The arm was extended on the experimental platform for electrical signal testing. This enabled the monitoring of the pulse.

## 3. Results

### 3.1. Fiber Modification

Figure 2a–d depicts the untreated electrospun fiber membrane alongside the morphological alterations on the fiber’s surface following its immersion in a dopamine solution for varying durations. In Figure 2a, the surface of the untreated electrospun fiber appears smooth. However, Figure 2b–d elucidate the morphological configuration of the fiber network and individual fibers subsequent to dopamine modification. Post 3 h of treatment, the fiber network’s surface evolves to a rougher texture, with the emergence of minute particles, as discerned through SEM analysis. This phenomenon arises from the adhesive polymerization of dopamine, initiating its adherence to the fiber surface. With prolonged immersion in the dopamine solution, a 6 h treatment prompts further transformation in the microscopic morphology of the fiber surface. As depicted in Figure 2c, in contrast to the 3 h treatment, certain polydopamine particles on the fiber surface exhibit a notable increase in diameter. This escalation stems from the ongoing polymerization process wherein fiber surface domains containing polydopamine particles, driven by functional groups, attract newly polymerized dopamine, thereby engendering polydopamine particles with a larger diameter. Following a 9 h immersion, as illustrated in Figure 2d, dopamine accumulation commences on the fiber surface, with select regions enveloped by a contiguous layer of polydopamine. The influence of polydopamine on the fiber is perceptible through the fiber network’s color alteration. Untreated PF presents as white, whereas immersion in the dopamine solution induces color transformation. Dopamine polymerization in a solution possessing a pH of 8.5 yields a brown hue, with the degree of brownness on the fiber indicative of the loaded polydopamine content. As depicted in Appendix A, untreated PF exhibits a white hue, progressively deepening in color with dopamine treatment and mirroring the augmentation in polydopamine content on the fiber’s surface, as observed in the electron microscope images over time.

Figure 2e presents the infiltration test images of PF, DPF-3, DPF-6, and DPF-9. On the PF sample, after the droplet falls, a significantly large contact angle is observed at the moment of contact with the fiber network. In the next 180 s, there is no significant change in the contact angle, indicating the hydrophobic nature of PF. This phenomenon may be related to the fiber material and fiber network structure. By contrast, samples DPF-3, DPF-6, and DPF-9 exhibit significantly enhanced hydrophilicity. The fiber network is completely infiltrated in a short period, indicating that dopamine modification makes the fiber surface hydrophilic. With the increase in surface polydopamine content, the infiltration angle decreases when the droplet contacts the sample, and the complete infiltration time gradually shortens. Dopamine treatment enhances the hydrophilicity of the fiber network, facilitating better binding with the hydrogel precursor. This helps prevent internal bubbles in the hydrogel due to infiltration issues, thereby improving the performance of the DPF hydrogel sensor. Dopamine modification improves this material’s affinity and adhesion by introducing dopamine molecules to the material surface. In this process, dopamine, as an “intermediate molecule”, can bind with the fiber through its special adhesive properties. Due to the numerous active groups in dopamine, it can form intermolecular forces with large molecular chains and water molecules, facilitating the formation of an interface between the fiber and the hydrogel. This further enhances its mechanical properties and optimizes internal ion channels.

By selecting the reinforcing material for the DPF hydrogel, in addition to the infiltration performance, we also considered the impact of polydopamine on the fiber during the preparation process. We composite the gel with fiber, and after 40 min of heat curing at 90 °C, hydrogels with different dopamine contents exhibit varying degrees of solidification on the fiber (Appendix A). Considering that DPF-3, DPF-6, and DPF-9 all show good infiltration performance when the polymerization time in dopamine is too long, the hydrogel is difficult to solidify. Therefore, we chose hydrogels with a polymerization time of 3 h as the experimental material.

### 3.2. Mechanical Testing

Figure 2f shows the stress–strain curves of the hydrogel and DPF hydrogel. Robust mechanical performance is paramount for the sensing capabilities of flexible sensors, necessitating a thorough analysis of these mechanical properties. As depicted in the graph, the tensile fracture length of the DPF hydrogel sensor attains 640%, accompanied by a maximum tensile stress exceeding 700 kPa. In contrast, conventional hydrogel achieves a tensile fracture length of up to 1300%, albeit with a maximum tensile stress of approximately 90 kPa. With the introduction of fibers as reinforcing elements, the tensile fracture elongation of the DPF hydrogel experiences a slight reduction, yet its deformation range still reaches 640%, adequately meeting the strain requirements of flexible sensors. Flexible sensors, characterized as elastic materials, undergo both elastic and plastic deformation when subjected to external forces. Elastic deformation denotes the material’s capacity to swiftly revert to its original state after stress, while plastic deformation involves maintaining its shape without succumbing to a permanent alteration. To align with usage prerequisites, we prioritized elastic deformation in flexible sensors. The modulus of elasticity directly governs both elastic and plastic deformation, epitomizing the slope of the stress–strain relationship and indicating the material’s resilience against external forces. Materials boasting a higher modulus of elasticity typically manifest increased rigidity and elastic recovery, signifying their ability to promptly revert to their original state post-stress.

The control group hydrogel, characterized by its diminished strength, is susceptible to plastic deformation post-stress, rendering the restoration to its initial state challenging. Consequently, there is a decline in the stability of the sensing performance of flexible sensors. Conversely, the integration of fibers in the DPF hydrogel, coupled with the interface effect forged between fibers and the hydrogel, markedly amplifies the modulus of elasticity and maximum stress of the flexible sensor. This imparts superior elastic recovery to the DPF hydrogel. Owing to the composite structure, the forces applied to the hydrogel are transmitted from the interface to the fiber, mitigating the occurrence of crack expansion phenomena. Additionally, the DPF hydrogel demonstrates a notable enhancement in tensile fracture work in comparison to the hydrogel. The formula for the fracture energy of the hydrogel is given by the fracture work equal to the area under the stress–strain curve as follows:$$W=\;\int_{0}^{\Delta l}F(l)dl;$$
where *W* is the fracture energy, *F*(*l*) is the stress–strain curve, and Δ*l* is elongation at the applied stress. Following fiber composite material modification, the fracture work of the sensor elevates from 35 kJ to an impressive 203 kJ, reflecting a 5.6-fold improvement.

The DPF hydrogel, functioning as an elastic entity, contends with a more intricate force milieu, which is subject to substantial deformations in contrast to conventional sensors. Mechanical hysteresis, with its potential to impede response speed, accuracy, and real-time efficacy in flexible sensors, becomes critical for consideration. This phenomenon induces a delay in the stabilization of the output signal following external pressure or deformation, particularly pronounced in materials with significant mechanical hysteresis. Yet, the integration of fibers serves to augment the modulus of the flexible sensor as a heightened modulus, aiding in the amelioration of mechanical hysteresis. As delineated in Figure 2g, we conducted stretching examinations on the DPF hydrogel, imposing strains of 10%, 50%, and 100%, and meticulously documenting the cyclic stress–strain curves throughout the complete stretching cycle. By scrutinizing the loading and unloading curves within the stress–strain relationship, we discerned the presence of mechanical hysteresis. Under conditions of minimal deformation, the loading curve of the DPF hydrogel is closely aligned with the unloading curve, predominantly manifesting elastic deformation akin to that of an elastic substance, with negligible hysteresis effects. This underscores the exceptional elastic attributes of the DPF hydrogel under modest deformations, effectively mitigating the potential repercussions of mechanical hysteresis on sensor performance.

### 3.3. Sensing Performance

Figure 3a portrays the outcomes of stretching the DPF hydrogel and the control group hydrogel at strain levels of 10%, 20%, 30%, 40%, 50%, and 70%. The comparative analysis of the relative resistance change rates of the two sensor groups across five stretching cycles reveals valuable insights. The examination of six datasets demonstrates that, irrespective of the DPF hydrogel or hydrogel, the relative resistance change rates progressively escalate with deformation, underscoring the effective perception of external mechanical signals by both sensor groups. Notably, when subjected to identical mechanical stimuli, the relative resistance change rate of the DPF hydrogel surpasses that of the hydrogel, highlighting the ability of the composite structure to augment the sensitivity of flexible sensors. This enhancement stems from the increased mobilization of the hydrogel’s three-dimensional molecular chain skeleton and alterations in ion transport rates induced by the internal fibers and interfaces of the flexible sensor under external forces, thereby heightening the sensitivity of the hydrogel sensor.

The stretching of the DPF hydrogel sensor until fracture ensues and the relative resistance change in the flexible sensor throughout this process is meticulously documented. Illustrated in Figure 3b, this curve delineates the progression. The sensor’s detection range surpasses 600%, indicative of its expansive detection capabilities. Should the detection range of a flexible sensor fail to encompass the variation range of specific application scenarios, it risks providing inaccurate and unreliable data. A broader detection range enhances the adaptability of flexible sensors, enabling them to accommodate a wider array of environmental conditions and object characteristics.

The responsiveness of a sensor denotes its capacity to promptly react to alterations in external stimuli, which is a pivotal attribute in myriad applications. This rapidity significantly impacts the efficacy and utility of flexible sensors in specific environments or applications. In certain scenarios, the swift and precise responsiveness of sensors to external changes stands as a critical factor in ensuring data precision and accuracy. A heightened response speed ensures that the data collected by the sensor closely mirrors actual changes. Figure 3c illustrates the response speed assessment of the flexible sensor, showcasing an impressive response speed of up to 50 milliseconds. Furthermore, upon the cessation of external force, the flexible sensor swiftly reverts to its original state within a mere 100 milliseconds. This attests to the sensor’s remarkable response speed, coupled with its high elastic modulus, contributing to a rapid recovery time and thereby fully aligning with the requisites of diverse usage scenarios. Appendix A delineates the trapezoidal tensile testing model of the sensor. This testing modality involves cyclic testing with intervals between motion and rest, systematically recording the electrical signal changes during the stretching process at 10 s intervals facilitated by the stretching machine. As evident in the Appendix A, the sensor adeptly maintains stability for the sensing performance during both motion and rest, with no discernible occurrence of electrical signal drift. This underscores its suitability for detecting human motion and other physiological signals.

Cycle stability, denoting the capability of a flexible sensor to uphold performance and response characteristics over repeated use or cycles, significantly influences the stability and dependability of flexible sensors, which are imperative for sustained application. Figure 3d presents the stability examination of the DPF hydrogel sensor, showcasing impeccable electrical stability even after enduring 2000 continuous cycles of stretching. Within the cycle stability assessment, as delineated by the magnified segment in Figure 3d (cycles 400–405 and 1400–1405), the sensor’s signal exhibits unwavering stability. The DPF sensor as a composite structure utilizes modified fiber as a toughening material, exhibiting an interface interaction between the fiber and the hydrogel. Consequently, when subjected to external forces, the sensor transfers the force from the matrix (hydrogel) to the reinforcement (fiber), preventing the breakdown of the internal macromolecular framework of the hydrogel. Moreover, the internal fiber further inhibits crack propagation. This sensor demonstrates favorable performance in practical use. Through comprehensive testing encompassing sensitivity, detection range, response speed, and stability, the DPF hydrogel manifests outstanding performance.

### 3.4. Applications

We successfully engineered flexible sensors using DPF hydrogel, endowed with exemplary mechanical and electrical properties, thereby establishing an optimal material substratum for the construction of intelligent wearable devices. These sensors find application in crafting wearable devices that are designed to monitor various human physiological signals, as shown in Figure 4a. Capitalizing on their remarkable flexibility and elasticity, these sensors seamlessly conform to bodily contours and movements, ensuring the comfort of wearable devices. Their heightened sensitivity, rapid responsivity, and augmented stability further fortify the pragmatic utility of the sensor. Human physiological signals span a diverse range of movements, encompassing heartbeat pulses to joint motion, representing a substantial signal spectrum. Our sensor, distinguished by elevated sensitivity and adherence, proficiently monitors an array of nuanced signals. Employing refined testing methodologies, we successfully employed the DPF hydrogel for pulse signal monitoring (Figure 4b). The sensor exhibits notable elasticity and an extensive detection range, adeptly capturing even substantial joint movements, such as joint bending. Through the integration of the processed DPF hydrogel with a superelastic wristband, we achieved the monitoring of wrist joint bending (as shown in Figure 4c). The DPF hydrogel not only lends itself to the creation of flexible wearable devices for monitoring human physiological signals but also lends its application to the construction of flexible electronic devices. Through the astute amalgamation of conventional fabric, DPF hydrogel, and conductive fabric, we conceived a flexible electronic keyboard (refer to Appendix A for the keyboard structure). This keyboard employs the DPF hydrogel as a piezoresistive element, with the resistance altering upon the depression of a specific key. The resultant electrical signal is subsequently transmuted into a control signal for electronic devices. Figure 4d vividly illustrates our utilization of the prepared flexible keyboard to adeptly control a computer, steering a small ball through a maze. This sequence of innovative applications unequivocally underscores the vast potential of DPF hydrogel in the realms of wearable devices and flexible electronic apparatus, heralding new frontiers in related technological advancements.

Owing to the imperative for wearable devices to establish contact with the human body, their safety profoundly influences their practical application. This sensor employs the polyacrylamide hydrogel, which is suitable for applications in the medical and health domains, demonstrating notable compatibility and yielding no adverse impact on human health. Furthermore, the hydrogel employed in the fabrication process consumes minimal energy, possesses an extended operational lifespan, and exhibits characteristics of biodegradability and recyclability, aligning seamlessly with environmental requisites. These attributes confer a substantial potential value upon the DPF hydrogel sensor in pragmatic applications.

Nevertheless, for the tangible production and utilization of the sensor, certain challenges warrant attention. For instance, the electrostatic spinning method employed in fiber preparation faces setbacks in realizing its large-scale production. Moreover, the hydrogel sensor encounters difficulties in discerning diverse directions of mechanical stimuli within intricate environments. Additionally, given its hydrogel nature, its electrical properties may exhibit drift with fluctuations in temperature. Real-world applications pose these challenges to the hydrogel flexible sensor. Mitigating these issues remains a prospective avenue for future research concerning hydrogel flexible sensors.

## 4. Conclusions

In this study, we designed a composite hydrogel sensor by integrating modified elastic fibers and a hydrogel to augment the interfacial synergy between the components and refine the properties of the hydrogel. We selected modified fibers as a reinforcement, enhancing the mechanical performance of the sensor and endowing remarkable sensing capabilities to the DPF hydrogel. Relative to pure hydrogel, the fiber-composite hydrogel demonstrated markedly heightened sensitivity. The incorporation of fibers not only elevated the sensor’s fracture energy from 35 kJ to 203 kJ but also extended the detection range to 640%. Its electrical attributes remained stable over 2000 test cycles, with a response time of only 50 ms. Consequently, this sensor displays considerable promise for use in applications such as electronic skin, human health monitoring, and motion detection. This study describes a potential strategy for use in amplifying the sensing and mechanical attributes of hydrogel sensors via composite modifications, enabling innovative research and applications in related fields.

## Figures and Tables

**Figure 1 nanomaterials-14-00438-f001:**
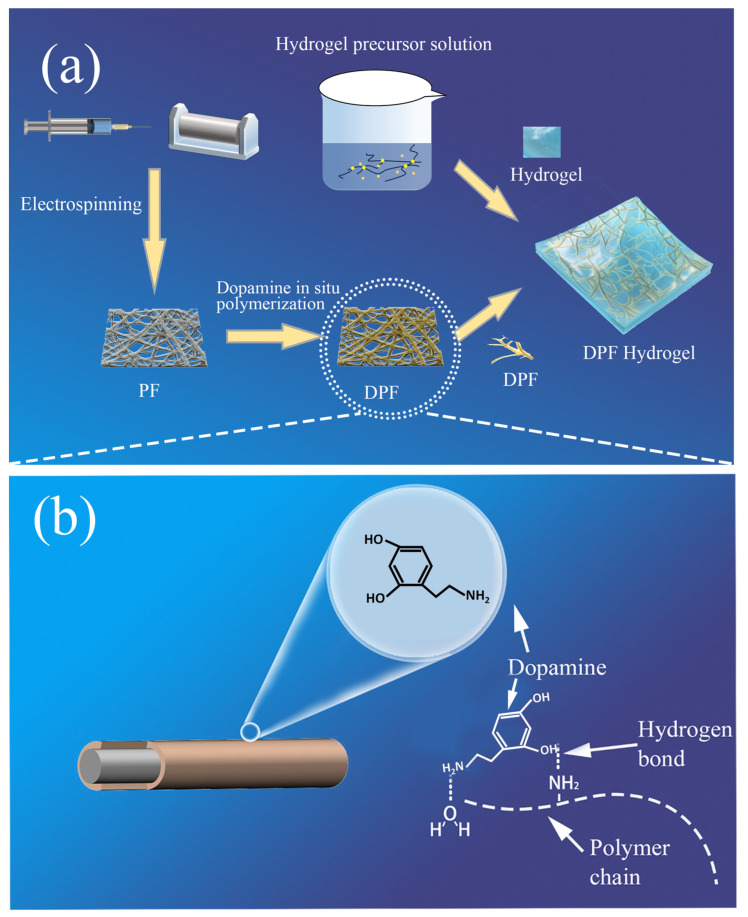
(**a**) Schematic illustration of DPF hydrogel preparation process; (**b**) schematic diagram of dopamine-modified polyurethane fiber (DPF) and its functional groups.

**Figure 2 nanomaterials-14-00438-f002:**
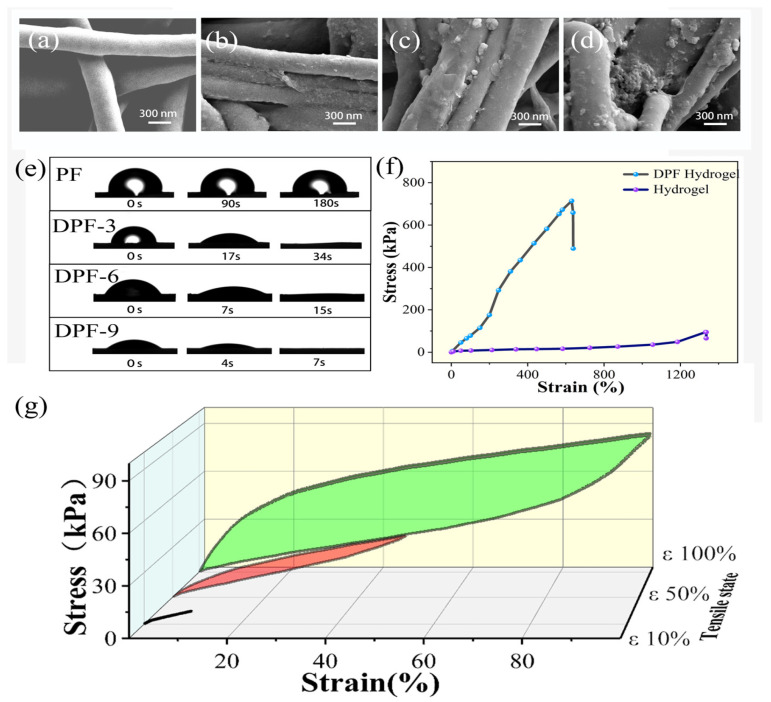
(**a**–**d**) SEM images of PF, DPF-3, DPF-6, and DPF-9; (**e**) Infiltration testing of samples PF, DPF-3, DPF-6, and DPF-9; (**f**) Stress–strain curves of hydrogel and the DPF hydrogel; (**g**) Cyclic stress–strain curves of the DPF hydrogel sensor at strains of 10%, 50%, and 100%.

**Figure 3 nanomaterials-14-00438-f003:**
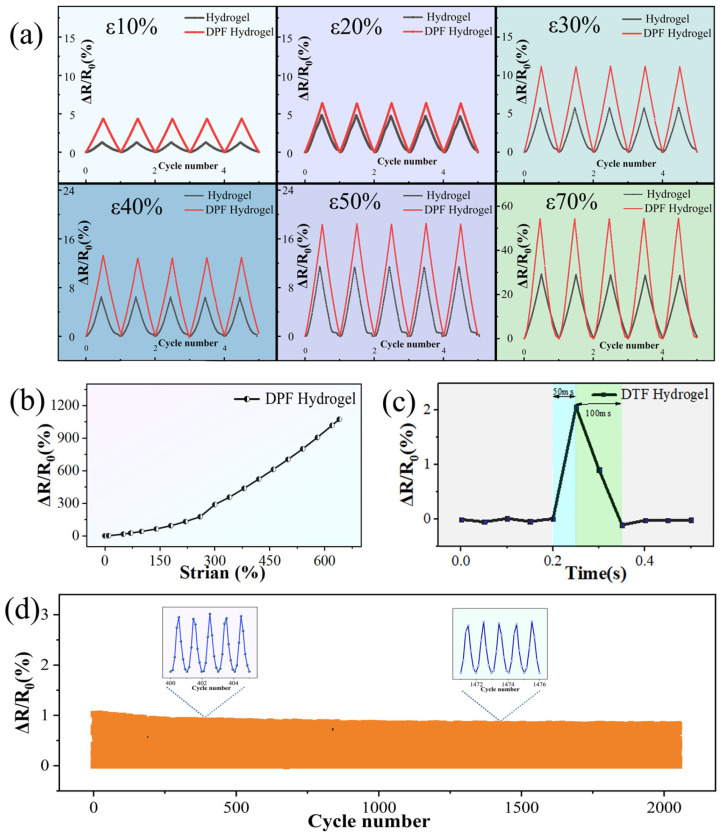
(**a**) Comparison of responsiveness between DPF hydrogel and hydrogel; (**b**) responsiveness of DPF hydrogel when subjected to draw fracture (0–640%); (**c**) sensor response speed of DPF hydrogel; and (**d**) DPF hydrogel tested at 10% elongation for 2000 draw cycles.

**Figure 4 nanomaterials-14-00438-f004:**
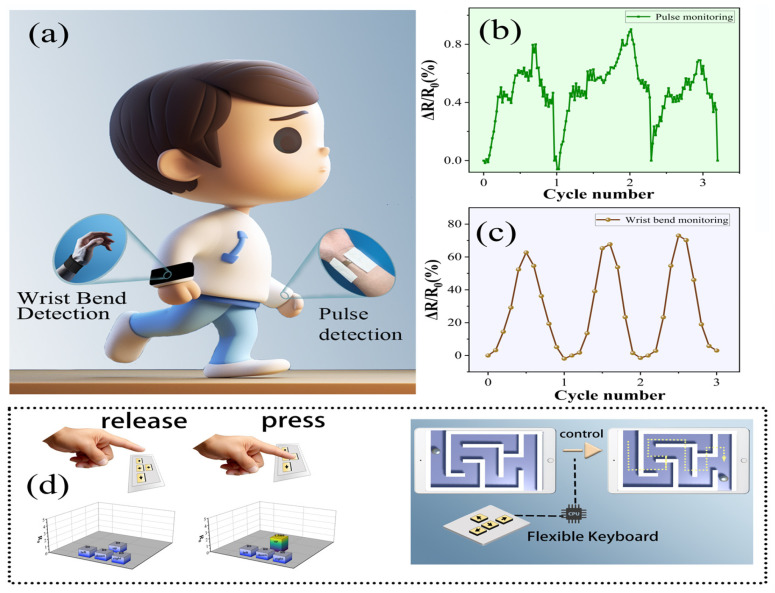
(**a**) Sensors for use in surveilling human physiological signals; (**b**) wearable sensors for use in monitoring pulse rates; and (**c**) wearable sensors for use in monitoring wrist flexion. (**d**) Integration of the DPF hydrogel in fabricating a flexible keyboard and its functional demonstration.

## Data Availability

The data presented in this study are available on request from the co-responding author.

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
