# Peer review of "Electronic Skin Based on Polydopamine-Modified Superelastic Fibers with Superior Conductivity and Durability"

_nanomaterials, 2024, doi:10.3390/nano14050438_

Round 1

Reviewer 1 Report

Comments and Suggestions for Authors

The manuscript describes the use of polydopamine-modified electrospun fibers to enhance the mechanical properties of ionic hydrogels. The ionic hydrogels and their various modification methods with electrospun fibers are studied in the literature with a recent review paper by Zhang et al. The dopamine modification of nanofibers is also the subject of extensive research.  The authors do not seem to be aware of the existing research in this field. The manuscript organization and detail level lack what is expected from a scientific manuscript. The study has to be compared with the existing studies and its novelty should be clearly explained. Here are some of my specific comments;

1)  The materials and methods chapter refers to the supporting information for the description of DTF synthesis in the first sentence. I believe this is important and details have to be in the main text. However more importantly the Supp. information has no detail about the synthesis. 

2) The abbreviations TF and DTF do not make sense. What does T stand for?

3) The Figure 1 organization color scheme has to be improved. The 1a is difficult to read. 1b is not informative. 1c has a different color than the rest etc.

3) Fig 2e should be labeled to show which panel corresponds to which composite. Fig 2b and c do not look different in terms of dopamine; why do they behave so differently?

4) Fig S1 is claimed to corroborate with the increased adherence of dopamine to the fiber surface. However, this is not explained well. Is the change in the color indication of dopamine, why?

5) It is very hard to read Fig 2f and 2g.  What does the depth axis show in 2g?

6) The pulse monitoring data is not convincing. How is it measured, how can we be sure it is coming from the pulse and not a movement artifact? What is the cycle correspond to? The figure legend pulse monitoring is not appropriate.

7) Fig 3b should read Strain in x axis.

Author Response

We thank you very much for giving us an opportunity to revise our manuscript, we appreciate editor and reviewers very much for their valuable and very helpful comments and suggestions on our manuscript entitled “Electronic Skin based on polydopamine modified superelastic fibers with superior conductivity and durability”. We have studied comments carefully and have modified the manuscript accordingly. We have placed the modified content and details in the attachment, hoping that you are satisfied with our revisions.

Reviewer 2 Report

Comments and Suggestions for Authors

11.       While the electrical properties seem stable after 2000 cycles of testing, the long-term stability of the composite hydrogel needs to be thoroughly examined. How well does it hold up over an extended period of use?

22.       The study mentions the promise of intelligent wearable devices, but the practical application of the composite hydrogel in real-world scenarios needs to be explored. Are there any limitations or factors that may affect its performance in different conditions?

33.       The method for crafting the composite-structured hydrogel should be scalable and cost-effective for practical implementation. Are there any challenges in mass production, and how feasible is the manufacturing process?

44.       Since the hydrogel is intended for use in wearable devices, its interaction with the human body needs thorough evaluation. Is the composite hydrogel biocompatible, and does it pose any risks or reactions when in contact with the skin over an extended period?

55.       Considering the increasing focus on sustainability, it would be beneficial to assess the environmental impact of the materials used in the composite hydrogel. Are the components environmentally friendly, and what is the overall ecological footprint of the manufacturing process?

Comments on the Quality of English Language

as above.

Author Response

(The authors gave the same response as above.)

Round 2

Reviewer 1 Report

Comments and Suggestions for Authors

The revisions are mostly satisfactory and the manuscript can be published after some revisions. 

1) Figure 2g, the depth axis is missing

2) The new abbreviations (DPF instead of DTF) are not adapted everywhere. Supp. info. is still referring to the structures as DTF and in conclusion it is referred to as PDF.

3) The detailed explanations such as 1) the red and orange rectangles explaining some of the features and 2) the details of the pulse measurement are useful and can be adapted in the manuscript to help the readers. 

Author Response

We appreciate editor and reviewers very much for their valuable and very helpful comments and suggestions on our manuscript entitled “Electronic Skin based on polydopamine modified superelastic fibers with superior conductivity and durability”. We have studied comments carefully and have modified the manuscript accordingly. We have placed the modified content and details in the attachment, hoping that you are satisfied with our revisions.

Reviewer 2 Report

Comments and Suggestions for Authors

Please include the answers to the questions inside the manuscript. 

Comments on the Quality of English Language

As above.

Author Response

(The authors gave the same response as above.)
